# Making the Decision to Stay at Home: Developing a Community-Based Care Process Model for Aging in Place

**DOI:** 10.3390/ijerph18115987

**Published:** 2021-06-02

**Authors:** Katarina Galof, Zvone Balantič

**Affiliations:** 1Department of Occupational Therapy, Faculty of Health Sciences, University of Ljubljana, 1000 Ljubljana, Slovenia; 2Faculty of Organizational Sciences, University of Maribor, 4000 Kranj, Slovenia; zvone.balantic@um.si

**Keywords:** independent functioning of older adults, home and community-based providers, health care, framework, spiral model

## Abstract

The care of older adults who wish to spend their old age at home should be regulated in every country. The purpose of this article is to illustrate the steps for developing a community-based care process model (CBCPM), applied to a real-world phenomenon, using an inductive, theory-generative research approach to enable aging at home. The contribution to practice is that the collaboration team experts facilitate the application of the process in their own work as non-professional human resources. This means that each older adult is his or her own case study. Different experts and non-experts can engage in the process of meeting needs as required. The empirical work examined the number of levels and steps required and the types of human resources needed. The proposed typology of the CBCPM for older adults can provide insight, offer a useful framework for future policy development, and evaluate pilots at a time when this area of legislation is being implemented.

## 1. Introduction

The same situation as that in Europe exists in America with a growing population of older adults, where findings have shown that exemptions for older adults are primarily focused on helping people in their own homes to age in place [1]. Older people have the right to protection from poverty and social exclusion, which can be achieved through intergenerational integration. It is important to enable older people to live healthy, safe and contented lives [2].

In 2019, about 16.5% of the U.S. population [3] and 20.3% of the European Union population [4] were 65 or older. In the European Union, this was 0.3% more than the year before and 2.9% more than the corresponding proportion from the decade before [4,5,6]. In Slovenia, the share of older adults aged 65 years and over increased rapidly in 2020, accounting for 20.4% of the Slovenian population at the beginning of the year [7]; between 2010 and 2020, this share increased by 3.9% [7] and is expected to reach 30.7% by 2050 [4]. A similar situation regarding the increase of the elderly population by 2050 is expected with the European Union reaching 29.5% and in America reaching 22% [3]. In recent years, the aging population has come into focus as a cause for concern as it is expected that the way people work and retire will change in order to maintain themselves. If a population is expected to live longer than previous generations, the economy must also change to be able, and be required, to meet the needs of citizens. Therefore, contributions to the integration of older adults into society will become increasingly important.

By the end of 2020, pilot projects were implemented in Slovenia at the national level as part of the design of the long-term care system. Policy-makers wanted to review the foundations established so far, especially when entering the system or assessing the needs of individuals, and when determining the entry threshold for potential recipients of long-term care services. As an example of good practice, Slovenia used the German model of public programs, which provides universal support for the cost of long-term services and basic support to almost all Germans with its self-funding, social insurance-like approach [8,9]. To determine the entry threshold of potential users, the German tool NBA-SLO [10] was adapted and validated for the Slovenian setting. Therefore, the focus of the state pilot was on determining the entry threshold.

Occupations are everyday activities people do as individuals and with communities to occupy time and bring meaning and purpose to life [11]. Activities of daily living-ADL [12] are oriented towards taking care of their own bodies and are completed on a routine basis. Instrumental activities of daily living-IADL [13] support daily life within the home and community. To perform ADL and IADL, older adults also need performance skills (motor skills, process skills and social interaction skills). Skills are supported by the context in which the performance occurs, including environmental and older adult factors [14].

The development of the community-based care process model (CBCPM) for older adults was based on the integration and implementation of daily activities in the home setting to maintain their autonomy and independence in carrying out daily activities [15,16]. The client-centered approach [17] was used to design the process model of community-based care and its integral components. The client-centered approach is partnership, empowerment, engagement, participation and negotiation, all processes supported by active listening to, respect for and meeting the needs of older adults to enable them to make informed decisions about their care [18]. They participated in decision-making, were involved in the implementation of activities, and in the selection of actors who assisted them in the implementation of a particular activity with which they had a personal relationship, regardless of their profession.

The theoretical frameworks of the models of the different health disciplines [19,20] contributed to the implementation of the activities of the older adults. Environmental factors also play an important role [21] in the development of the CBCPM because people with low incomes, multiple chronic illnesses, or limited social supports, such as those who live in unsafe neighborhoods or poor-quality housing do not receive effective health care [22].

Care for older adults who wish to spend their twilight years at home is often based on informal, usually unpaid, care provided by family members, friends, or neighbors who require assistance due to health or functional needs [23]. Evidence [24] suggests that disabled and older adults tend not to want institutional care, and that families and other informal caregivers strongly prefer to continue caring for their family members in need of care in a friendly environment, such as at home and in local communities [23,25,26]. At present, and in the future, policy makers see home-based care as a sustainable approach to prevent the need for unnecessary acute and/or long-term institutionalization and to keep people in their homes and communities for as long as possible [24].

The purpose of this article is to illustrate the stages of model development, applied to a real life phenomenon, using an inductive, theory-generating research approach to enable older adults in Slovenia to age at home. The contribution to practice is to enable collaboration with human resources. The teamwork process in the model connects health and social professionals in their work (formal caregivers) as non-professional human resources (informal caregivers). Formal caregivers are professionals with unique competencies and who have had formal training. The field of formal caregivers in Slovenia is under the Ministry of Health and the Ministry of Labor, Family, Social Affairs and Equal Opportunities. In order to solve the health problems of older adults, the health professionals available are nursing assistants, practical nurses, registered nurses, occupational therapists and physiotherapists. On the other hand, there are also social care professions such as personal assistants, social caregivers, social workers and social gerontologists who are responsible for solving social care problems. The field of informal caregivers is very large, regardless of the regional or national level. Informal caregivers are members of pensioners associations, the Red Cross, Caritas and many other governmental and non-governmental organizations. Sometimes they are simply family members or neighbors with no association or affiliation. This means that each older adult is their own case study, where different experts as formal caregivers and volunteers as informal caregivers can enter the process of meeting the needs of older adults when they are needed, regardless of the level of the model or the steps of the model.

Research Question:

What type of human resources in the community-based care process model do we need to enable older adults aging in their home environment to meet their needs, be as active as possible, and to improve or maintain quality of life levels?

## 2. Materials and Methods

We chose the spiral model methodology [27,28,29] because it is a good option for large and complex projects, such as the one we have in long-term care. The progressive nature of the spiral model allows us to break down a large project into smaller pieces (formal and informal human resources) and address one feature at a time (enabling older adults to perform their daily activities and occupations) to ensure nothing is overlooked (personal factors, contexts, performance patterns, older adults’ performance abilities). In addition, because the prototype is built incrementally, it is sometimes easier to estimate the cost of the entire project.

We used inductive reasoning [30] for detailed observations of the world of older adults. We then moved on to abstract generalizations and ideas about ageing at home. Once we had taken an inductive approach starting with a topic, we tended to develop empirical generalizations and make tentative connections as we progressed in the research. In the early stages of the research, hypotheses may not have been found because we used an inductive research approach and we were unsure of the nature and type of research findings until the study was complete.

In developing the CBCPM in the home setting, we researched areas on micro and macro ergonomics [31,32], individual elements of the International Classification of Functioning, Disability and Health (ICF) [19], occupational therapy models such as person, environment occupational performance model [33,34] and occupational therapy practice framework [20]. Knowledge of project management and human resource management [35] was also needed to design a process model of community care to achieve the goal of enabling older people to age at home.

The basic steps for model creation or model construction (in model building) are the same for all modelling methods. The details vary somewhat from method to method, but an understanding of the common steps, combined with the typical underlying assumptions required for analysis, provides a framework within which the results from almost any method can be interpreted and understood [36].

The four basic phase steps (P_1_—identify, P_2_—evaluate, P_3_—develop, P_4_—plan) used to develop the CBCPM are part of Boehm’s spiral model [27]. We expanded the model development with the time component and two phase steps on the level [37]. Finally, we obtained the six phase steps (P_1_ to P_6_) at each level (I to IV) with the time dimension (Figure 1). Understanding the common model, combined with the typical basic assumptions necessary to analyze and meet the needs of older adults, provides a framework for older adults living in the community.

The spiral model methodology with a temporal dimension allows and demonstrates clearer monitoring of the development of human resources to enable the individual needs of an older adult in the long-term care system to be met. With completion at each level or stage on the abscissa axis, as we work through all the basic phase steps and the spiral describes a 360° angle, we move to the next level on the time axis. The levels (I to IV) follow one another, from concept development (level I) to new product development (level II) to product growth (level III) and finally to product maintenance (level IV).

Thus, when treating individual older adults, we adhere to four basic levels of the spiral [27,28] and six basic phase steps [29,38]. Through the project method of involving different stakeholders in the implementation of care services, the product life cycles were interwoven at each level.

We involved each stakeholder according to their professional or technical competencies in the development and implementation of care services, thus avoiding mutual overlap of interests.

When searching for answers, we decided to use a refined and modified six characteristic basic phase steps (P_1_ to P_6_) spiral model, which is shown in the matrix distribution in Figure 1 and which Balantič [39] used when introducing e-learning materials in medicine.

Human resource analysis was combined with individual elements of modern ergonomics [40,41,42] and understanding the needs of older adults [1,16,43] and used to design a CBCPM. Human resources were integrated into the long-term care system incrementally or at the level of an organized unit in the form of an interdisciplinary team.

## 3. Results

The following summarizes the concept of the developed cyclical change introduction to the older adults’ care system in the form of a CBCPM.

### 3.1. Levels (I to IV) and Characteristic Steps (P_1 to_ P_6_) of CBCPM

An illustration of the Level Cycles and Phase Steps of the CBCPM compared to the basic Boehm model with four basic phase steps (P_1_—identify, P_2_—evaluate, P_3_—develop, P_4_—plan) is shown above (Figure 1). The steps of each level of the CBCPM are repeated in a specific order and according to the competencies of the stakeholders involved in the process. The modified spiral model of care for older adults retains four levels (I to IV) and six basic phase steps (A to F) of each cycle, which are described in more detail below (Figure 2).

### 3.2. Level I—Concept Development Project

The reason for the first level step was the initiative of the older adults (P_1.1_) This initiative has long been heard from older adults, but recently it has become even louder. It is an initiative to spend the autumn of their lives in a home-like living environment. For this kind of living, the older adults need help in meeting their needs from the point of view of independent living and participating in the performance of daily activities in a home living environment. In order to realize the presented idea, the expert team needs representatives of the Ministries of Health, Labor and Education, representatives of the Institute of Social Welfare, representatives of higher educational institutions in the field of health and social care (e.g., faculties of health, faculties of social work), representatives of the Association of Occupational Therapists, Chamber of Nursing, Association of Physiotherapists, Social Chamber), representatives of the Center of Social Work and representatives of the creators of national professional qualifications. Next, it is the task of the representative team to prepare an assessment of the initiative to meet the needs related to the independent living of older adults and their participation in the performance and implementation of daily activities in the home living environment (P_1.2_). The results of the initial analysis show that it is necessary to involve experts in the field of service providers and in the field of training of stakeholders for service delivery.

The risk analysis (P_1.3_) in the development phase of the Long-Term Care Workforce Plan emphasizes the evaluation of emerging alternatives for multi-stakeholder engagement and the identification of risks in decision-making. The plan also provides support and assistance to educational institutions in creating awareness of the importance of engaging older adults in satisfying and performing daily activities in the home environment.

The outcome of the project work will provide technical solutions (P_1.4_) for the implementation of human resources to enable older people to perform daily activities in the home living environment. Additional SWOT analysis will improve the prototype version of the solutions of the proposed human resources. This will be followed by the selection of appropriate personnel (P_1.5_) and their release. The final group of stakeholders involved is expected to include nursing, occupational therapy, physiotherapy, social work and social care. The final step of the first level (P_1.6_) will be the testing of the set of all stakeholders involved among potential users, which will set the stage for moving to the second level.

### 3.3. Level II—New Product Development Project

The team, consisting of stakeholders from the health, social and educational sectors (identical to the first level), selects the test results of the participating stakeholders identified as the end-user’s condition in the last step of the first level. (P_2.1_). Long-term care human resource development system planning also provides a schema or plan for implementing stakeholder involvement in long-term care (P_2.2_). In the implementation plan, the key elements will be the time and staffing standards of the service provided. A description of general and specific competencies will also be part of the implementation plan. The risk analysis (P_2.3_) will be identical in content to the analysis from the first level, except that it will provide us with additional content coverage of the amount of human resources. In the fourth step of the second level, we will formulate a final proposal for the set of stakeholders involved (P_2.4_). The set of stakeholders involved and the level of independence or need for assistance in performing daily activities in the home living environment will be reviewed among older adults living at home. We will conduct a survey (P_2.5_). Based on the results obtained, we will assess the set of stakeholders involved (P_2.6_), which is extremely important for the transition to the third level.

### 3.4. Level III—Product Growth Project

At the product growth planning level, communication with users of long-term care is required (P_3.1_). The central importance of the second step (P_3.2_) will be to identify the need for education and communication with service users, which will be reflected in the formation of the research question (independence in performing daily activities) and the third hypothesis (actors involved in service delivery). Therefore, we expect a low level of risk analysis regarding product growth (P_3.3_). In step four, we will improve and complete the development of the stakeholder set (P_3.4_). Thus, we will arrive at the final formation of a stakeholder set for the delivery of long-term care services in the home environment (P_3.5_). At this stage, in the final step (P_3.6_), we will look at planning and developing further alternatives for long-term care. We will focus our considerations on the development of a possible upgrade of the existing human resources proposal and the involvement of new stakeholders in the process of implementing long-term care.

### 3.5. Level IV—Product Maintenance Project

It will also be necessary to communicate with users at the last level (P_4.1_). In certain places it will be necessary to inform the public about the available set of actors as well as the available set of services for care receivers. Certain places could be described as entry points into the care system. We see this opportunity as existing in facilities such as a health center, a social work center, a home help facility. We have reached the point where we will be able to assess the applicability of the proposed solution to the staffing system in long-term care (P_4.2_). The next step will be to confirm that the human resources in health and social care with the profiles of the proposed group of actors meet the needs of long-term care (P_4.3_). The analysis will lead us to the introduction of systemic solutions in the field of training as well as in the field of competencies of each stakeholder. With the steps (P_4.4_) and (P_4.5_), we will further intervene in the area of introduction of systemic solutions at the country level. The proposal to involve the actors in the long-term care system (such as occupation-al therapists, physiotherapists, community nurses, social workers and social assistants) will be presented to policy makers and stimulated for reflection before the Long-Term Care Act (P_4.6_) is passed and the proposed systemic solutions are implemented.

## 4. Discussion

The answer to the research question is that we need formal and informal caregivers to enable older adults to age at home. The proposed idea helps us to realize the development of the presented CBCPM, whose implementation has five reasons.

The first reason why we need a CBCPM is based on the recommendations of WHO [24] that older adults prefer to continue caring for their family members in a friendly environment where it is not enough that we only have informal caregivers who are mostly not professionally trained [16,26,44,45], but in most cases we also need formal caregivers who are trained and competent.

The second reason is that the proportion of older adults in the population is increasing both globally and in Slovenia, and only 5% of Slovenian older adults are in institutional care [46], but it is possible that these older adults have a lower value of support received than non-institutionalized older adults [44]. Addressing the shortage of direct care workers requires a multi-faceted approach that includes better wages, benefits, and education and training programs to draw people into the workforce [45]. Thus, for the realization of new jobs in the home environment, it is necessary to unite and select health and social care professionals, depending on the auspices under which the ministries fall and who funds them. The CBCPM, in accordance with the competencies of formal caregivers, enables the independent operation of older adults in their home environment, and their inclusion according to individual needs requires teamwork.

The third reason is based on the fact that aging in place is not yet regulated in Slovenia, which means that those who live at home have to rely on their own resourcefulness when they need services to live independently. This may be considered a violation of human principles or the basic law of human rights to ensure a decent life for all citizens.

The fourth reason is that with the adaptation of the German tool NBA-SLO, we have an assessment basis that will be reflected in the assessment modules for entry into the long-term care system. The NBA-SLO assessment tool will facilitate our work and help us decide which of the formal or/and informal caregivers should be included in the newly developed CBCPM for home treatment of older adults.

The last reason is that we do not have national tools to identify the needs of older adults to create a plan for aging in place when they need more support, as we found for Americans PlanYourLifespan.org (PYL) [47]. We can use the stages of model development to implement older adults’ activities by including environmental factors as part of the context. In the future, this will also be a way to develop software to implement the CBCPM in practice.

Now, combining all of the above reasons, we require a multidisciplinary type of care for older adults in the home setting. One of the very important considerations is the formal stakeholders, which means that we treat older adults from all possible aspects and with all available formal and informal human resources to jointly achieve the set goal. The proposed interdisciplinary CBCPM allows for the involvement of stakeholders at each level (group work of specialists), with each individual carrying out their work and discussing the results as a team. We believe that in order to successfully implement the proposed model in practice, it is crucial to educate the primary care physician about the potential range of different stakeholders and their professional competencies that can help older adults to live better and more independently with their knowledge, advice and treatment.

Municipalities, as one of the possible bearers of the financial burden of care services, have a number of available stakeholders from interdisciplinary teams (health and social) and governmental and non-governmental organizations at the municipal level. For this purpose or for the needs of care of older adults, it is also possible to expand the patronage activity in health centers. As part of the primary health care activity, it is possible to create a multidisciplinary team of stakeholders in collaboration with existing community social work centers and home help providers, with other stakeholders from the governmental and nongovernmental sectors participating alongside the multidisciplinary team. As another option for the bearer of the financial burden, we see an alternative in various forms of supplementary health insurance. Klančar, Švab and Kersnik [48] believe that the size of the facility should be adapted to the local environment and the capabilities of the services for the rational provision of primary health care. If we consider the mentioned alternative ways of treating older adults in the home environment, besides the impact on the improvement of the quality of life itself, we also contribute to the fulfillment of their greatest desire to spend the autumn of their lives in the safe shelter of their own home. This is supported by the findings of Iwarsson, Horstmann, Carlsson, Oswald, and Wahl [42], in which evaluations for environmental interventions based on inclusion in the human-environment model assessment were found to be more effective than those that considered only environmental factors as inclusion criteria. For example, physical, mental and psychological impairments are assessed in the recording and evaluation process in the German NBA SLO tool. The assessment is used to measure a person’s degree of independence in six different areas (mobility, cognitive abilities, behavior and psychological problems, ADLs, coping and independent handling of illness—or therapy-related demands and burdens, organization of everyday life and social contacts, activities outside the home, and household management), which are then combined—on the basis of different weightings—into an overall score [10].

Stakeholders (from health or social care) whose life goals are interwoven through individual-level services are organized and networked through project management. By repeating phases and improving processes, each individual stakeholder forms a spiral as they enter the process of improving older adults’ independence in enabling aging at home, the six steps from A to F (A—communication with potential users, B—planning, C—risk analysis, D—design, E—transfer to practice, and F- evaluation) at each level from I to IV (I—concept development project, II—new product development project, III—product growth project, and IV—product maintenance project). Looking at Figure 2, the CBCPM looks like a spiral with many loops. The number of loops varies depending on the needs of the older adult and is often determined by the professional skills of those involved. Each loop of the spiral in the long-term care development process is one of the repeating phases of the CBCPM. Each stakeholder is able to repeat treatment cycles to achieve a higher level of independence in the performance of daily activities in older adults.

Home care is a very complex issue that needs to be effective. Older adults have their own desires about who can provide support or assistance in their home environment [26]. We believe that the introduction of the process model (CBCPM) is a possible way to address the health and social problems of home care for older adults from the perspective of an interdisciplinary team. By linking informal home care providers with the interdisciplinary team, we obtain a multidisciplinary team that is concerned with maintaining the independence of older adults in performing daily activities, depending on their needs and current psychophysical health status. Thus, an integral part of the treatment of older adults in the home living environment will be based on the professional skills of the stakeholders of the multidisciplinary team involved in the CBCPM. In this type of treatment, in addition to curative activities, we aim to provide preventive guidance to older adults in the functional design of the home environment and to provide appropriate support and training to family members. We consider individual elements of the ergonomic system and the use of assistive technologies (medical devices, electronic systems to control the home environment, and communication technologies). Ergonomics with three typical categories such as physical, cognitive and organizational is related to productivity and activities [32].

When implementing a CBCPM in practice, errors that occur at the stakeholder level and the collaboration with the interdisciplinary team can be quickly identified. Since the process model provides the opportunity to repeat individual phases and thus improve processes, any errors that occur can be corrected and eliminated according to protocol. The purpose of implementing a process model is so that when potential errors are identified, such as at the stakeholder level due to noncompliance with procedures, time standards, political will, and human resources, they are corrected and eliminated as the number of loops varies depending on the needs of the older adult. The CBCPM provides the ability to repeat individual phase steps to improve processes.

In order to achieve the set goals of Ageing in Place for users and to apply the CBCPM with emphasis on the elements of cognitive ergonomics in practice, each stakeholder of the multidisciplinary team will create their own spiral in the implementation of long-term care, presenting their own project management. Opportunities to engage formal and informal caregivers depend on their availability, the education system, and policies at the national level, and vary from country to country. It is important that we focus on the resources available and familiarize ourselves with the professional skills that enable us to enable older adults to live in the home environment. It is also important to pass on the knowledge and training to the informal team members who will care for the older adults. By gradually involving the available team members, we will treat the older adults as a whole and as a team, gradually enabling them to maintain independent performance of daily activities. For example, the physical therapist will increase hip joint mobility after hip surgery, the occupational therapist will use appropriate strategies to assist them to put on their pants after hip surgery, the family members will provide an appropriate chair (appropriate sitting height) for the older adult to rest during the day. There are three different goals in this example and we have three cases of projects that develop and grow with a time dimension. To achieve the example goal, each formal caregiver must take six steps within four stages as outlined by the CBCPM concept (Figure 2). The CBCPM allows the life cycles of services to be intertwined and stakeholders to be involved at each stage according to the needs of the users or their professional competence. The individual stages of the spiral model represent the loop of the service’s own development process. The activities of the intermediate stages of the spiral model are repeated at each level of the spiral model. The presented spiral grows with the time dimension as a treatment process that improves the older adult’s ability to function and become or remain independent in the activities of daily living. This means that the spiral grows or increases from phase to phase steps and from level to level. Because the radius of the spiral is the cost of the project, formal caregivers in particular must adhere to the code of ethics when providing services. This is an interesting topic for further discussion on what the economic impact of implementing CBCPM might look like in practice.

## 5. Conclusions

For this reason, we have developed a human services model, CBCPM, to address older adults living in the home environment. The model is based on the integration and implementation of the daily activities of the older adults in the home environment. The presented CBCPM for older adults includes formal home health providers, informal home care providers, and policy. It is an affordable and flexible approach to the needs of older adults, allowing them to spend their old age in their own environment and helping to postpone the transition to institutional care. In this article, the authors highlight and present the developmental stages of the process model. Its application at the level of the primary health care system and at the community level will contribute significantly to the quality of independent living of older adults in the home environment.

One of the solutions to Aging in Place lies in the implementation of a spiral model (CBCPM) for the treatment of older adults in their home environment involving citizens, formal and informal care providers and policy makers. We believe that this type of treatment will be more acceptable to older adults because it is based on the legal foundation of social and health legislation and allows access to a quality services in the home environment through various human resources.

For Aging in Place, it is crucial to determine not only the threshold for entry and the associated economic impact on the government’s budget, insurance companies and individuals, but also to include the human resources that will provide services in the long-term care system. If we are to put the CBCPM model into practice to meet the needs of older adults when they are needed, we must first develop a system of services for each formal caregiver involved according to their professional and technical competencies.

## Figures and Tables

**Figure 1 ijerph-18-05987-f001:**
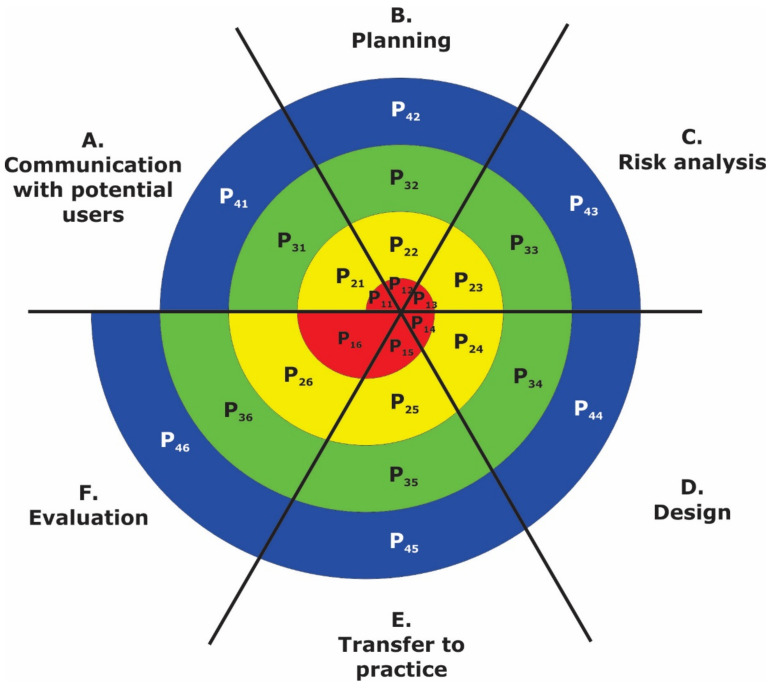
Matrix characteristic basic phase steps (P1 to P_6_ in CBCPM renamed as (**A**–**F**)), (**A**)—communication with potential users (P_1_), (**B**)—planning (P_2_), (**C**)—risk analysis (P_3_), (**D**)—design (P_4_), (**E**)—transfer to practice (P_5_), (**F**)—evaluation (P_6_).

**Figure 2 ijerph-18-05987-f002:**
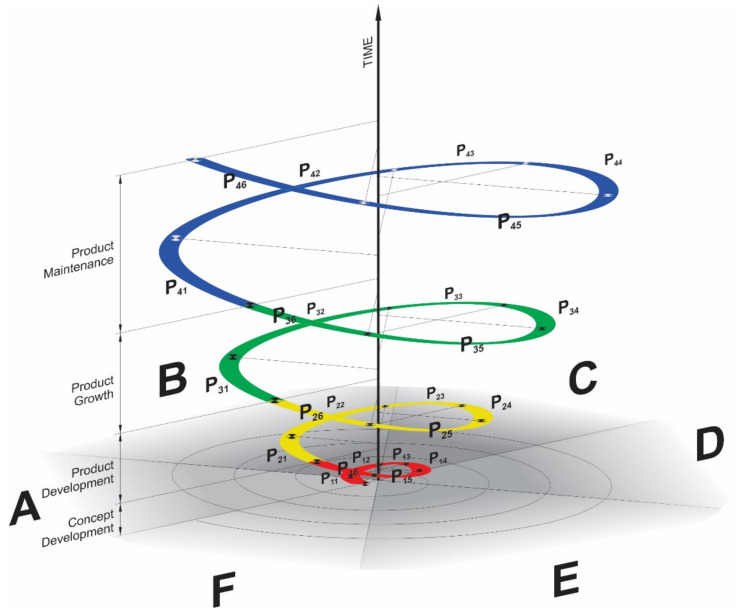
Characteristic six basic phase steps (**A**–**F**) and levels (I to IV) of the CBCPM, (**A**)—communication with potential users, (**B**)—planning, (**C**)—risk analysis, (**D**)—design, (**E**)—transfer to practice, (**F**)–evaluation, I—concept development project, II—new product development project, III—product growth project, IV—product maintenance project.

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
