# Peer review of "Making the Decision to Stay at Home: Developing a Community-Based Care Process Model for Aging in Place"

_ijerph, 2021, doi:10.3390/ijerph18115987_

Round 1
Reviewer 1 Report
Dear Editor,
Thank you very much for the opportunity to review this manuscript.
The manuscript titled Making the decision to stay at home: Developing a community-based care process model for Aging in Place aimed to answer the research question: “What type of human resources in the community based care process model do we need to enable older adults aging in their home environment to meet their needs, be as active as possible, and improve or maintain quality of life levels?”
I have made some specific comments as follows:
Introduction
The authors have conducted a comprehensive review on the growing population of older adults and the expectation of aging in place from older adults and carers. The reason why knowing the type of human resources in the process model has yet provided in the background. The authors are recommended to highlight the needs of different experts in the care model so as to link up with the research question.
The paragraph (line 84-90) described about the methodology which seems more appropriate to be placed in the “Materials and methods” section.
Materials and methods
For the description of the spiral model in line 145 -149 using Pij is rather confusing. It will be more clear to reader to specify the meaning of the number 1-4 and 1-6 in the matrix distribution. The authors need to revise this paragraph to make it understandable.
Results
The “four basic field steps (identify, evaluate, develop, plan)” mentioned in line 166 have not been mentioned in the method section, it seems to be confused with the matrix characteristic steps: A - communication with potential users, B - planning, C 158 - risk analysis, D - design, E - transfer to practice, F- evaluation. In particular, it is not sure what the word “steps” in the last sentence of line 166 refers to.
Discussion
The first paragraph seems repeating and should be described in the introduction section.
More discussion on how the results can address the research question is needed.
Author Response
Response to Reviewer 1 Comments
Introduction
The paragraph (lines 97-107) highlights the various health and social care experts currently available nationally.
The paragraph (lines 84-90) describing the methodology has been moved to the Materials and Methods section.
Materials and methods
We revise Figure I and Figure II with the aim of making numbers 1-4 and 1-6 in the matrix distribution more understandable.
Results
In order to present the "four basic field steps (identify, evaluate, develop, plan)" mentioned in line 166 in a clearer way, we add their explanation in the "Methods" section starting from line 144.
Discussion
In the discussion, we linked the findings to the research questions. All five reasons for implementing the CBCPM model in practice are now more clearly explained. Despite the other five reasons for implementing the model in practice, it is also important to note that our baseline model includes formal and informal caregivers, depending on their availability at the national level. In the future, the implementation process could also be a new opportunity to develop new professions as needed.

Reviewer 2 Report
This paper address an underestimated field of health sciences, with an interesting methodology.
However, the readibility is a bit difficult, and I would suggest a moderate revision of the english language and style.
As the methods are probably the most interesting and innovative section of the paper, I would suggest to emphasize them, better describing the followed steps of the process.
Author Response
Response to Reviewer 2 Comments
The article has been revised by a native speaker with the aim of improving the English language and style.
We have improved the methods section with a new illustration to make it more specific and clearer so that it is easier to understand.
We have standardized the terminology related to each basic phase step and level, which is also reflected in the new illustrations. We describe the development process by modifying Boehm's four basic phase steps into six basic phase steps in the CBCPM and linking them to our new classification.
